# Transforming Generic Coder LLMs to Effective Binary Code Embedding Models for Similarity Detection

**Litao Li**[*]
Queen's University
litao.li@queensu.ca

**Leo Song**[*]
Queen's University
leo.song@queensu.ca

**Steven H. H. Ding**
McGill University
steven.h.ding@mcgill.ca

**Benjamin C.M. Fung**
McGill University
ben.fung@mcgill.ca

**Philippe Charland**
Mission Critical Cyber Security Section, Defence R&D Canada
philippe.charland@drdc-rddc.gc.ca

## Abstract

Cybersecurity and software research have crossed paths with modern deep learning research for a few years. The power of large language models (LLMs) in particular has intrigued us to apply them to understanding binary code. In this paper, we investigate some of the many ways LLMs can be applied to binary code similarity detection, as it is a significantly more difficult task compared to source code similarity detection due to the sparsity of information and less meaningful syntax. It also has great practical implications, such as vulnerability and malware detection. We find that pretrained LLMs are mostly capable of detecting similar binary code, even with a zero-shot setting. Our main contributions and findings are to provide several supervised fine-tuning methods that, when combined, significantly surpass zero-shot LLMs and state-of-the-art binary code similarity detection methods. Specifically, we up-train the model through data augmentation, translation-style causal learning, LLM2Vec, and cumulative GTE loss. With a complete ablation study, we show that our training method can transform a generic language model into a powerful binary similarity expert, and is also robust and general enough for cross-optimization, cross-architecture, and cross-obfuscation detection.

## 1 Introduction

LLMs are almost ubiquitous in everyday life, fueling many powerful applications. Cybersecurity and software researchers have utilized language models to solve complex tasks, such as code generation, summarization, and similarity detection. In this paper, we investigate the benefits that LLMs can bring to binary code modeling. In particular, binary code similarity detection (BCSD) and retrieval are difficult due to the sparsity of information, loss of semantics and structures, and its limited syntax compared to source code. Moreover, the same source code can be compiled into many formats of binary code using different compiler settings and environments, including optimization levels, machine architectures, compiler software, and obfuscation techniques. Existing LLMs [Roziere *et al.*, 2023; Radford, 2018] can somewhat perform binary code analysis with their large training data of trillions of natural language and code tokens, but the proportion of binary code is small. Downstream tasks like similarity detection and vulnerability detection are still difficult and require fine-tuning. Some open-source coder models like CodeT5 [Wang *et al.*, 2021] and GraphCodeBERT [Guo *et al.*, 2020] even lack the training of binary code, making zero-shot useless for binary analysis.

---

[*]Equal contribution.

39th Conference on Neural Information Processing Systems (NeurIPS 2025).

Many binary code models are trained from scratch [Yang *et al.*, 2021; Tian *et al.*, 2021; Yu *et al.*, 2020] and their sizes are much smaller compared to LLMs. They typically focus on cross-optimization retrieval and fail to generalize to diverse compiler settings. Other works take pre-trained LLMs [Tan *et al.*, 2024; Wang *et al.*, 2022, 2024] and apply custom fine-tuning techniques for binary code matching. However, most of these approaches rely on either closed-source models like GPT [Radford, 2018] or large model sizes. This leads to scalability issues when the computational resource is a constraint, which realistically is the case with small research labs or even companies. We want to explore effective and efficient LLM fine-tuning for binary code embedding and matching.

In this work, we address the aforementioned problems and propose **EBM** (**E**ffective **B**inary **M**atching), our novel training framework, including carefully chosen data augmentation and fine-tuning processes, to uptrain a generic LLM into a binary code embedding and matching expert. We show in our experiments that LLMs have become dominant enough that even zero-shot models can surpass well-trained binary code models. With our fine-tuning applied, EBM can significantly increase the mean reciprocal rank (MRR) by 10% to 70%, depending on the tasks. We also provide a comprehensive ablation study to prove and emphasize the importance of each training process. The dataset and code can be accessed on Github [1]. Our major contributions are:

- We propose a multi-training framework to utilize the power of generic LLMs specifically for BCSD. We target different compiler settings, including cross-optimization, cross-architecture, and cross-obfuscation, to build a general and effective similarity retriever.

- To combat the lack of assembly code LLMs, we uptrain the generic LLM to a binary code-specific model. Particularly in cross-architecture similarity detection, this allows for significantly better translation between different syntaxes.

- We utilize LLM2Vec to build a refinement of the assembly tokens, which encodes better semantics through a masked next token prediction task.

- In the downstream contrastive learning task, we propose an enhanced version of InfoNCE loss to utilize all available samples within the batch. This is particularly useful when computing resources are limited and large models are trained.

- We build and compare various baselines to evaluate the effect of different language/coder models and state-of-the-art BCSD models. In our similarity retrieval evaluation, our approach outperforms all benchmarks for both datasets.

- We conduct thorough ablation studies and in-depth analysis to investigate all our training tasks and how they contribute to the similarity retrieval result. We show that all our training tasks are essential and can improve retrieval performance.

## 2 Related Works

**Traditional BCSD** Without data-driven or learning-based methods, code similarity detection is traditionally conducted using static analysis, dynamic analysis, or code-based algorithms. Static analysis usually involves graph matching [Dullien and Rolles, 2005; Bourquin *et al.*, 2013], where control flow graphs are extracted from assembly code and compared using algorithms or user-defined heuristics. Dynamic analysis instead leverages runtime or symbolic execution to investigate program behavior [Pewny *et al.*, 2015; Xu *et al.*, 2020; Egele *et al.*, 2014; Moser *et al.*, 2007]. Execution traces or paths can be profiled for either manual or automated comparison using distance-based or statistics-based methods. Code-based algorithms rely on patterns of the actual binary code, which can be opcode or instruction code. Various ways of analyzing code strings include distance metrics (Smith-Waterman or Levenshtein distance) [Gao *et al.*, 2008], N-Gram matching [Rosenblum *et al.*, 2008], and frequency analysis [Santos *et al.*, 2013].

**Machine Learning-based BCSD** is effective when a large amount of training data is available. The methods can generally be categorized into text-based, structure-based, or combined. Text-based models treat assembly instructions or opcodes as tokens and feed them into various models such as unsupervised [Ding *et al.*, 2019], RNN-based [Massarelli *et al.*, 2019b; Yang *et al.*, 2021; Tian *et al.*, 2021], BERT-based [Koo *et al.*, 2021; Ahn *et al.*, 2022; Li *et al.*, 2023a], and LLM-based [Tan *et al.*, 2024; Wang *et al.*, 2022; Liu *et al.*, 2023; Wang *et al.*, 2024]. Such models focus on the semantics

---

[1]Github Link

and syntax of the binary code to encode binary programs. In structure-based models, some form of graph structures, such as abstract syntax trees, control flow graphs, or custom graphs, is parsed from binary code. Then appropriate machine learning techniques can be applied, including tree-LSTM [Tai *et al.*, 2015; Yang *et al.*, 2021] and graph neural network [Xu *et al.*, 2017; Li *et al.*, 2019]. Instead of utilizing complicated language models for word embedding, these models often apply Word2Vec-style [Mikolov *et al.*, 2013] representation learning to obtain node features. Other research works combine code and structure information into a streamlined learning process [Massarelli *et al.*, 2019a; Gao *et al.*, 2018; Yu *et al.*, 2020]. When training the networks, contrastive learning is the most popular technique for supervised learning models to differentiate positive and negative samples, which are fed into the network as a pair of input data. Many approaches also utilize a siamese architecture to simultaneously encode pairs under the same set of model parameters. Recently, many similarity detection methods have adopted the InfoNCE loss [Oord *et al.*, 2018] to enhance the discriminative power of negative pairs by maximizing the mutual information. It has been shown to be quite effective and can sometimes lead to large performance gains, especially for LLMs where batch size tends to be small. Readers may refer to Appendix. A for additional information about LLM architecture and foundation models.

## 3 Methodologies

We define the task of BCSD as retrieving the most similar function from a pool of binary functions. Similar binary functions are created from the same source code with different compiler settings. In our case, the settings can be optimization-based, obfuscation-based, architecture-based, and compiler-based. The input is a binary function $x$ and the output is a list of functions, ranked by cosine similarity. For evaluation purposes, the pool contains exactly one similar function. The complete training undergoes four processes: data augmentation, translation learning, embedding training, and contrastive learning.

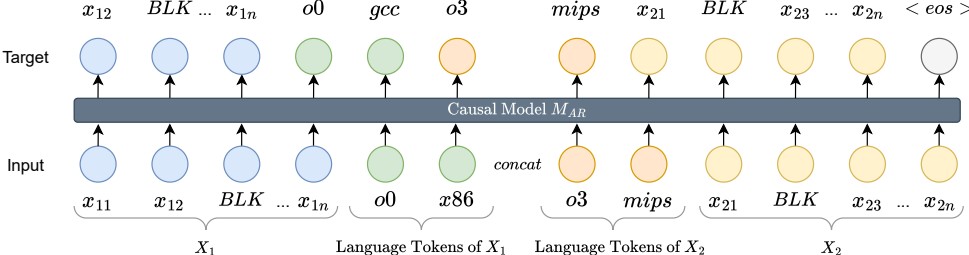

Figure 1: The input and target of causal uptraining are shown here. The input contains sampled pairs of semantically identical assembly functions, where **BLK** tokens are added for structure awareness, and language tokens are added for translation awareness. The training follows an autoregressive manner, where previous tokens are used to predict the next unseen token.

### 3.1 Enabling Translation and Structure Awareness

Assembly code is an intermediate code acquired from a disassembler, such as IDAPRO [2]. To clean the data, all addresses, strings, and bytes are replaced by special tokens: **addr, byte, str**. The special tokens help reduce noise when addresses or bytes are arbitrary and also reduce context length after tokenization. We flatten the original data structure of assembly code, which contains basic blocks and instruction streams, into a single sentence. Doing this removes the requirement for hierarchical architecture and layered attention to process the inputs. This allows us to directly use existing pre-trained language and coder models. The drawback is that a single sentence no longer contains structural information regarding the semantically bound sections of the code (i.e., basic blocks). We propose to simply add a special **BLK** token in between basic blocks to increase structure awareness during training. This enables the model to consider assembly structure, which can be easily learned in our uptraining processes.

---

[2] https://hex-rays.com/ida-pro

Cross-architecture similarity detection is a difficult task that, to our knowledge, has not been extensively studied along with other compiler settings like cross-optimization. Assembly code can have a completely different syntax based on the machine architecture during compilation. Different architectures can be seen as different "languages" in this context. Inspired by [Conneau and Lample, 2019], we add several additional tokens to indicate the "language" information of each binary function, which corresponds to the optimization level, compiler, obfuscation, and architecture. Adding such information has proven to be effective, especially for cross-architecture detection.

### 3.2 Binary Translation Continual Training

Modern coder models aim to assist humans in coding tasks like code generation and summarization. Although assembly code is included in some of the existing models, it is not as well understood to the extent of source code languages like Python or Java. A cheap and effective way to uptrain such coder models with assembly code is to use an autoregressive setup, which trains the model to predict the next tokens.

We treat the task similarly to translation in natural language, as different compilers produce various syntaxes depending on the settings. This is especially true in cross-architecture similarity detection, where it is often not possible to retrieve similar code between machines with architectures like x86 and PowerPC. We only sample semantically identical function pairs $(X_1, X_2)$ and concatenate them into a single sequence $X = concat(X_1, X_2)$ as input to the model. A pre-trained coder AR model is initialized with weights $W^0$ and fine-tuned with the causal objective and update rule:

$$\mathcal{L}_{\mathcal{AR}} = -\sum_{i=1}^{n} \log P(x_i \mid x_{<i}) \tag{1}$$

$$W^{(t+1)} = W^{(t)} - \eta \cdot \nabla_W \mathcal{L}_{\mathcal{AR}}(X; W^{(t)}) \tag{2}$$

Where $\eta$ is the learning rate. As discussed in section 3.1, we manually add "language" tokens to all assembly functions as an indication of their compiler settings. For training the translation task, we place these tokens in between the concatenation of function pairs, acting as a transition. We give an illustration of the input and output format in Figure 1. It is important to let the model access these tokens during training before predicting the second sentence, $X_2$, as it provides generalization due to knowing what settings to expect and predict. It is important to reiterate that these language tokens are unavailable during the inference phase.

### 3.3 LLM2Vec

BERT [Devlin, 2018] models have been proven effective in embedding tasks, largely due to their bidirectional attention mechanism, which enables them to capture contextual information from the whole sequence [Reimers, 2019; Shi *et al.*, 2023]. However, the landscape of NLP has shifted significantly in recent years to LLMs trained with next token prediction, such as GPT-2 [Radford *et al.*, 2019], Llama [Touvron *et al.*, 2023], and their successors. These models, typically based on unidirectional architectures, have achieved state-of-the-art performance across a wide range of tasks, including embedding tasks, often surpassing traditional bidirectional models in scalability and generalization [Wang *et al.*, 2023a].

$$\mathcal{L}_{\text{MNTP}} = -\sum_{i}^{masked} P(x_i) \log P(\hat{x}_i | x_1, ..., x_n) \tag{3}$$

To bridge the gap between the strengths of causal models and the benefits of bidirectional architectures, we adopt an innovative approach called LLM2Vec [BehnamGhader *et al.*, 2024]. As depicted in Figure 2, this method leverages the pre-trained knowledge of causal models, while enabling bidirectional attention and being trained through a masked next token prediction task (MNTP). Specifically, we initialize the model with the same weights from a pre-trained causal model and enable bidirectional attention during forward propagation. The model is then fine-tuned using MNTP loss, defined in Equation 3, where a random token in the sequence is masked and predicted based on the surrounding context.

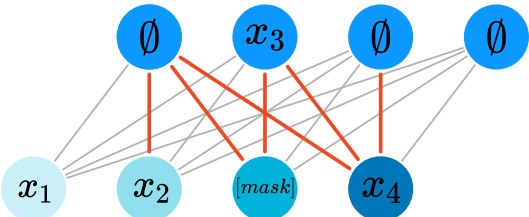

Figure 2: An overview of the LLM2Vec approach. The method adds bidirectional attention (red lines) based on future context to the original causal model (gray lines). It is fine-tuned with masked next token prediction (MNTP). The model can be further fine-tuned with contrastive learning approaches such as InfoNCE and GTE losses, for specific downstream tasks.

In essence, this approach constructs a BERT-like model from a pre-trained causal model rather than training a BERT model from scratch. This strategy offers several key advantages. First, it allows us to leverage the existing pre-trained LLMs, reducing the need for costly and time-consuming pre-training. Most importantly, it achieves better performance than training a BERT-style model from scratch.

### 3.4 Cumulative GTE Loss

The InfoNCE loss (Equation 4) is a widely used objective function in contrastive learning. It encourages the model to learn distinctive representations by contrasting one positive pair against in-batch negative ones. Incorporating additional contrastive pairs can further enhance the learning process [Li *et al.*, 2023b; Wang *et al.*, 2022]. We denote $k_i$ to be the embedding of the source binary function (key) in batch $i$, $q_i$ to be the embedding of the target binary function (query) in batch $i$, and $\tau$ as a hyperparameter for temperature.

$$\mathcal{L}_{\text{InfoNCE}} = -\frac{1}{N} \sum_{i=1}^{N} \log \frac{\exp(\text{sim}(q_i, k_i)/\tau)}{\sum_{j=1}^{N} \exp(\text{sim}(q_i, k_j)/\tau)} \tag{4}$$

The GTE (general text embedding) loss generalizes the InfoNCE loss by introducing additional contrastive terms, which include query-key pairs as well as query-query, key-key, and key-query pairs. This broader set of contrasts helps the model capture richer relationships within the data. The GTE loss is defined as:

$$\mathcal{L}_{\text{GTE}} = -\frac{1}{N} \sum_{i=1}^{N} \log \frac{\exp(\text{sim}(q_i, k_i)/\tau)}{Z}$$
$$Z = \sum_{j} e^{\text{sim}(q_i, k_j)/\tau} + \sum_{j} e^{\text{sim}(q_i, q_j)/\tau}$$
$$+ \sum_{j \neq i} e^{\text{sim}(q_j, k_i)/\tau} + \sum_{j \neq i} e^{\text{sim}(k_j, k_i)/\tau} \tag{5}$$

Based on the GTE loss, we propose the cumulative GTE loss (cGTE), which aggregates embedding representations from multiple distributed models and accumulated inputs. The cGTE loss is formulated as:

$$q = q^{(1)}||q^{(2)}||...||q^{(n)}$$
$$k = k^{(1)}||k^{(2)}||...||k^{(n)}$$
$$\mathcal{L}_{\text{cGTE}} = \text{GTE}(q, k) \tag{6}$$

where the superscript denotes accumulated vector representation from distributed models or batches. The backpropagation is halted until a number of inputs are met. The cGTE loss syncs gradient backpropagation generated from asynchronous input batches and distributed models to the same model weights. cGTE creates more contrastive pairs with limited computing resources.

## 4  Training and Experiments Setup

### 4.1  Backbone Model and Baselines

We use Qwen2.5-Coder-0.5B [Hui *et al.*, 2024] as the backbone model for all training processes. We compare against existing non-LLM methods, including PalmTree [Li *et al.*, 2021], SAFE [Massarelli *et al.*, 2019b], OrderMatters [Yu *et al.*, 2020], and Asm2Vec [Ding *et al.*, 2019]. We also evaluate several state-of-the-art large coder models, such as GraphCodeBERT [Guo *et al.*, 2020], CodeT5+ [Wang *et al.*, 2023b], Qwen2.5-Coder-1.5B [Hui *et al.*, 2024], Qwen3 [Yang *et al.*, 2025], and CodeGemma [Team *et al.*, 2024]. All baseline models, except for Qwen2.5-Emb, are fine-tuned on the same training set using contrastive learning. Qwen2.5-Emb, being a pre-trained embedding model, is included as a baseline to indicate the differences between programming language code and binary code. As our model is trained on Qwen2.5-Coder-0.5B, the Qwen2.5-Coder-1.5B baseline is a good candidate for an ablation study.

### 4.2  Dataset

We use two datasets for training and evaluation. The first contains multiple libraries written in C and is used by other existing works [Ding *et al.*, 2019]. We manually compile each library using different optimization levels (O0, O1, O2, and O3), compilers (GCC and Clang), architectures (x86, PowerPC, Arm, and MIPS), and obfuscations (none, substitution, flatten, bogus control flow, and all). The training libraries used include BusyBox, Coreutils, Curl, ImageMagick, PuTTY, and SQLite. The evaluation libraries are GMP, LibTomCrypt, and OpenSSL. We separate the libraries for the purpose of out-of-domain evaluation, which is a common practice in binary code retrieval. The pool size for the retrieval evaluation is 1,000. The second dataset is BinaryCorp [Wang *et al.*, 2022], which is constructed based on ArchLinux[3] and Arch User Repository[4]. BinaryCorp only contains cross-optimization functions. We use the same training and testing splits and pool size of 10,000 as the original paper for comparison.

### 4.3  Training Details

We train our models on a machine with 4 NVIDIA RTX 6000 GPUs (24 GB). Each computing card can contain one batch of 4 entries with 512 tokens as the sequence length. We use AdamW as the training optimizer and a learning rate of $10^{-5}$. Every model is trained with 5 epochs and warmed up for $1000/[\text{batch size}]$ steps. The best model weights are based on the best validation contrastive loss.

## 5  Evaluation

In this section, we evaluate both datasets with LLM-based and non-LLM baselines. The evaluation metrics are mean reciprocal rank (MRR) and recall@1. Due to limited space and a large number of combinations for our evaluation setup, we only include a subset of the retrieval tasks. The remaining evaluation details, including additional tasks, training, and inference time, are in the Appendix B.

| Models | MRR | | | | | | Recall@1 | | | | | |
|---|---|---|---|---|---|---|---|---|---|---|---|---|
| | O0,O3 | O0,O1 | O0,O2 | O1,O3 | O2,O3 | Avg. | O0,O3 | O0,O1 | O0,O2 | O1,O3 | O2,O3 | Avg. |
| SAFE | 0.189 | 0.189 | 0.200 | 0.218 | 0.171 | 0.193 | 0.063 | 0.000 | 0.063 | 0.063 | 0.000 | 0.038 |
| PalmTree | 0.023 | 0.020 | 0.019 | 0.314 | 0.878 | 0.251 | 0.008 | 0.006 | 0.007 | 0.184 | 0.676 | 0.176 |
| Asm2Vec | 0.444 | 0.494 | 0.460 | 0.535 | 0.563 | 0.499 | 0.234 | 0.290 | 0.252 | 0.343 | 0.376 | 0.299 |
| OrderMatters | 0.006 | 0.006 | 0.008 | 0.006 | 0.006 | 0.006 | 0.000 | 0.001 | 0.002 | 0.001 | 0.000 | 0.001 |
| GraphCodeBERT (125M) | 0.636 | 0.757 | 0.673 | 0.792 | 0.920 | 0.756 | 0.560 | 0.694 | 0.602 | 0.722 | 0.895 | 0.695 |
| CodeT5+ (110M) | 0.604 | 0.650 | 0.629 | 0.830 | 0.893 | 0.721 | 0.532 | 0.572 | 0.552 | 0.783 | 0.869 | 0.662 |
| Qwen2.5-Emb (1.5B) | 0.569 | 0.648 | 0.573 | 0.773 | 0.907 | 0.694 | 0.498 | 0.578 | 0.505 | 0.699 | 0.875 | 0.631 |
| Qwen2.5-Coder (1.5B) | 0.758 | 0.881 | 0.807 | 0.864 | 0.936 | 0.849 | 0.706 | 0.842 | 0.757 | 0.810 | 0.912 | 0.805 |
| CodeGemma (2B) | 0.763 | 0.888 | 0.833 | 0.866 | 0.931 | 0.856 | 0.696 | 0.840 | 0.778 | 0.821 | 0.905 | 0.808 |
| **EBM (0.5B)** | **0.850** | **0.942** | **0.902** | **0.933** | **0.955** | **0.916** | **0.793** | **0.903** | **0.850** | **0.887** | **0.929** | **0.872** |

Table 1: Evaluation on cross-optimization settings (O0, O1, O2, and O3) with a pool size of 1,000.

---

[3]https://archlinux.org/packages/

[4]https://aur.archlinux.org/

## 5.1 Cross-optimization Evaluation

Table 1 highlights the performance comparison between various baselines and our approach for cross-optimization retrieval. LLMs, even with a zero-shot setting, can outperform existing non-LLM methods, which have been state-of-the-art in the past for binary similarity detection. However, they still struggle with more difficult retrieval tasks like **[O0, O3]**. EBM can consistently outperform other models across all optimization settings, achieving the highest MRR and Recall@1 scores, while also improving **[O0, O3]** significantly.

## 5.2 Cross-architecture

Cross-architecture has been shown to be the most difficult task for both LLMs and non-LLM baselines, as shown in Table 2. Even with contrastive fine-tuning, Qwen-2.5-Coder, which has 1.5 billion parameters, fails to effectively retrieve cross-architecture functions. Due to the difference in syntax, matching the semantics across several languages is nearly impossible with limited training. Our method outperforms all baselines by a large margin. This further proves the significance of the causal training process that contributes to the improvement in cross-architecture generalization. We will show firm results in ablation studies.

| Models | MRR | | | | Recall@1 | | | |
|---|---|---|---|---|---|---|---|---|
| | Arm, x64 | PowerPC, x64 | MIPS, x64 | Avg. | Arm, x64 | PowerPC, x64 | MIPS, x64 | Avg. |
| SAFE | 0.239 | 0.187 | 0.196 | 0.208 | 0.063 | 0.063 | 0.063 | 0.063 |
| PalmTree | 0.037 | 0.036 | 0.018 | 0.031 | 0.031 | 0.013 | 0.007 | 0.017 |
| Asm2Vec | 0.242 | 0.293 | 0.417 | 0.317 | 0.085 | 0.113 | 0.231 | 0.143 |
| OrderMatters | 0.007 | 0.007 | 0.007 | 0.007 | 0.002 | 0.000 | 0.001 | 0.001 |
| GraphCodeBERT (125M) | 0.067 | 0.269 | 0.495 | 0.277 | 0.037 | 0.204 | 0.419 | 0.220 |
| CodeT5+ (110M) | 0.056 | 0.303 | 0.462 | 0.274 | 0.035 | 0.227 | 0.392 | 0.218 |
| Qwen2.5-Emb (1.5B) | 0.039 | 0.059 | 0.409 | 0.169 | 0.031 | 0.035 | 0.331 | 0.132 |
| Qwen2.5-Coder (1.5B) | 0.256 | 0.481 | 0.548 | 0.428 | 0.179 | 0.380 | 0.442 | 0.334 |
| CodeGemma (2B) | 0.293 | 0.581 | 0.548 | 0.474 | 0.208 | 0.479 | 0.432 | 0.373 |
| **EBM (0.5B)** | **0.783** | **0.792** | **0.859** | **0.811** | **0.675** | **0.703** | **0.784** | **0.721** |

Table 2: Evaluation on cross-architecture settings (Arm, x86-64, PowerPC, and MIPS) with a pool size of 1,000.

## 5.3 Cross-obfuscation

Obfuscation techniques can introduce complex and confusing variants to a binary function and are considered the most difficult task for binary retrieval. In Table 3, all models have shown undesirable results for **[all, none]**, which retrieves vanilla functions from functions with all three obfuscation techniques applied. Our method still outperforms all other baselines. While investigating the functions in this task, we often found that the obfuscated function can be 10x as large as the vanilla function. With limited sequence length, this result is somewhat expected. We suspect a much longer sequence length can alleviate this constraint and improve performance.

## 5.4 BinaryCorp

Table 4 shows the evaluation of the BinaryCorp dataset. Among all benchmarks, jTrans [Wang *et al.*, 2022] and CLAP [Wang *et al.*, 2024] are recent LLM-based models for BCSD. They have shown great results compared to the other methods. CLAP has the closest performance compared to our method, though has a few drawbacks. Firstly, CLAP relies on GPT3.5, a closed-source model, to generate natural language explanations. It is also costly and difficult to scale the model. CLAP uses Llama 13B and 30B as the backbone model for supervised fine-tuning, which can become a constraint for resource-limited scenarios.

In conclusion, EBM is a general approach that can adapt any pre-trained coder model, such as the 0.5B model used in our study, for binary embedding matching tasks. This methodology eliminates the need for external expert models, such as GPT, or auxiliary inputs, such as control flow graphs. Thus, it simplifies the data pipeline and training process and makes it more accessible and easy to use.

| Models | MRR | | | | Recall@1 | | | |
|---|---|---|---|---|---|---|---|---|
| | all, none | none, bcf | sub, fla | Avg. | all, none | none, bcf | sub, fla | Avg. |
| SAFE | 0.256 | 0.181 | 0.264 | 0.234 | 0.0625 | 0.0625 | 0.125 | 0.083 |
| PalmTree | 0.122 | 0.289 | 0.215 | 0.209 | 0.060 | 0.200 | 0.083 | 0.114 |
| Asm2Vec | 0.200 | 0.181 | 0.264 | 0.215 | 0.069 | 0.063 | 0.125 | 0.086 |
| OrderMatters | 0.008 | 0.006 | 0.007 | 0.007 | 0.001 | 0.001 | 0.001 | 0.001 |
| GraphCodeBERT (125M) | 0.230 | 0.648 | 0.479 | 0.452 | 0.163 | 0.557 | 0.391 | 0.370 |
| CodeT5+ (110M) | 0.176 | 0.619 | 0.372 | 0.389 | 0.118 | 0.539 | 0.291 | 0.316 |
| Qwen2.5-Emb (1.5B) | 0.288 | 0.630 | 0.466 | 0.461 | 0.213 | 0.538 | 0.375 | 0.375 |
| Qwen2.5-Coder (1.5B) | 0.391 | 0.719 | 0.580 | 0.563 | 0.301 | 0.637 | 0.491 | 0.476 |
| CodeGemma (2B) | 0.454 | 0.796 | 0.571 | 0.607 | 0.356 | 0.735 | 0.473 | 0.521 |
| **EBM (0.5B)** | **0.531** | **0.815** | **0.784** | **0.710** | **0.454** | **0.738** | **0.713** | **0.635** |

Table 3: Evaluation on cross-obfuscation settings (all obfuscations, bogus control flow, flattened, and substitution) with a pool size of 1,000. The proposed EBM model outperforms all baselines by an absolute margin of over 15% in both MRR and Recall@1 metrics.

| Models | MRR | | | | | | Recall@1 | | | | | |
|---|---|---|---|---|---|---|---|---|---|---|---|---|
| | O0,O3 | O1,O3 | O2,O3 | O0,Os | O2,Os | **Avg.** | O0,O3 | O1,O3 | O2,O3 | O0,Os | O2,Os | **Avg.** |
| Gemini | 0.037 | 0.161 | 0.416 | 0.049 | 0.195 | 0.172 | 0.024 | 0.122 | 0.367 | 0.030 | 0.151 | 0.139 |
| SAFE | 0.127 | 0.345 | 0.643 | 0.147 | 0.377 | 0.328 | 0.068 | 0.247 | 0.575 | 0.079 | 0.283 | 0.250 |
| OrderMatters | 0.062 | 0.319 | 0.600 | 0.075 | 0.233 | 0.258 | 0.040 | 0.248 | 0.535 | 0.040 | 0.158 | 0.204 |
| Asm2Vec | 0.072 | 0.449 | 0.669 | 0.083 | 0.510 | 0.357 | 0.046 | 0.367 | 0.589 | 0.052 | 0.426 | 0.296 |
| PalmTree | 0.130 | 0.403 | 0.677 | 0.152 | 0.496 | 0.372 | 0.080 | 0.326 | 0.609 | 0.097 | 0.420 | 0.306 |
| jTrans (Zero Shot) | 0.137 | 0.490 | 0.693 | 0.182 | 0.513 | 0.403 | 0.088 | 0.412 | 0.622 | 0.122 | 0.430 | 0.335 |
| jTrans (Finetune) | 0.475 | 0.663 | 0.731 | 0.539 | 0.664 | 0.614 | 0.376 | 0.580 | 0.661 | 0.443 | 0.585 | 0.529 |
| CLAP | 0.764 | 0.903 | 0.941 | **0.813** | 0.877 | 0.860 | 0.719 | 0.875 | 0.920 | 0.774 | 0.847 | 0.827 |
| **EBM** | **0.779** | **0.911** | **0.955** | 0.808 | **0.909** | **0.872** | **0.725** | **0.882** | **0.942** | **0.808** | **0.889** | **0.849** |

Table 4: Evaluation on the BinaryCorp-3M dataset with pool size=10,000

## 5.5 Similarity Threshold

In the preceding sections, we assumed the presence of at least one matching pair in the search pool. However, in real-world scenarios, top-1 selection often yields high false positive rates. To address this limitation, we introduce a similarity threshold based on the distribution of similarity scores in the training data. Table 5 reports the mean and standard deviation for both similar and non-similar optimization pairs. We set the threshold at 'mean - std' for each model. The accuracy is then computed as the proportion of queries without matching pairs that correctly fall below this threshold, formulated as "# of queries with no matching pair / total number of test queries." EBM continues to outperform the best baseline model. The results for the remaining experiments are in the Appendix.

| Source | Dest | Accuracy | | Mean ± STD of Similar pairs | | Mean ± STD of Non-similar pairs | |
|---|---|---|---|---|---|---|---|
| | | CodeGemma | EBM | CodeGemma | EBM | CodeGemma | EBM |
| o0 | o1 | 0.472 | 0.719 | 0.904 ± 0.136 | 0.910 ± 0.106 | 0.207 ± 0.220 | 0.180 ± 0.180 |
| o0 | o2 | 0.387 | 0.676 | 0.868 ± 0.155 | 0.879 ± 0.120 | 0.208 ± 0.212 | 0.178 ± 0.172 |
| o0 | o3 | 0.236 | 0.458 | 0.830 ± 0.195 | 0.833 ± 0.173 | 0.210 ± 0.213 | 0.184 ± 0.174 |
| o1 | o2 | 0.675 | 0.831 | 0.937 ± 0.092 | 0.952 ± 0.076 | 0.176 ± 0.213 | 0.164 ± 0.173 |
| o1 | o3 | 0.438 | 0.675 | 0.899 ± 0.151 | 0.916 ± 0.134 | 0.168 ± 0.221 | 0.170 ± 0.176 |
| o2 | o3 | 0.618 | 0.793 | 0.955 ± 0.139 | 0.965 ± 0.133 | 0.183 ± 0.217 | 0.174 ± 0.173 |

Table 5: Model Accuracy for non-match and Similarity Metrics Comparison

## 5.6 Comparison of Base and EBM-trained Models

To further assess the effectiveness of EBM, we perform the same evaluation on the base model for similarity detection with Qwen2.5-0.5B and Qwen3-1.7B models to understand the difference between pre- and post-finetuning. Table. 6 illustrates the average MRR and Recall@1 across cross-optimization, cross-architecture, and cross-obfuscation. Without training, the base model struggles to

understand the semantics of assembly code, especially for cross-architecture and cross-obfuscation tasks. EBM clearly enables significant uplift for similarity detection and boosts the MRR up to 16X and Recall@1 up to 30X.

| Models | Average MRR | | | Average Recall@1 | | |
|---|---|---|---|---|---|---|
| | Cross-Optimization | Cross-Obfuscation | Cross-Architecture | Cross-Optimization | Cross-Obfuscation | Cross-Architecture |
| Qwen2.5-Coder-0.5B (Base) | 0.485 | 0.226 | 0.050 | 0.444 | 0.184 | 0.026 |
| Qwen2.5-Coder-0.5B (EBM) | 0.924 | 0.738 | 0.816 | 0.883 | 0.668 | 0.733 |
| Qwen3-1.7B (Base) | 0.284 | 0.023 | 0.028 | 0.228 | 0.011 | 0.015 |
| Qwen3-1.7B (EBM) | 0.917 | 0.700 | 0.826 | 0.884 | 0.627 | 0.761 |

Table 6: Evaluation on the Base Model and Finetuned Model

# 6  Ablation

We illustrate ablation studies that investigate the effects of data augmentation and fine-tuning processes. All parts of our method must be included, as they provide either generalization or significant performance improvement for a tuned LLM to understand the context, semantics, and structures of assembly code.

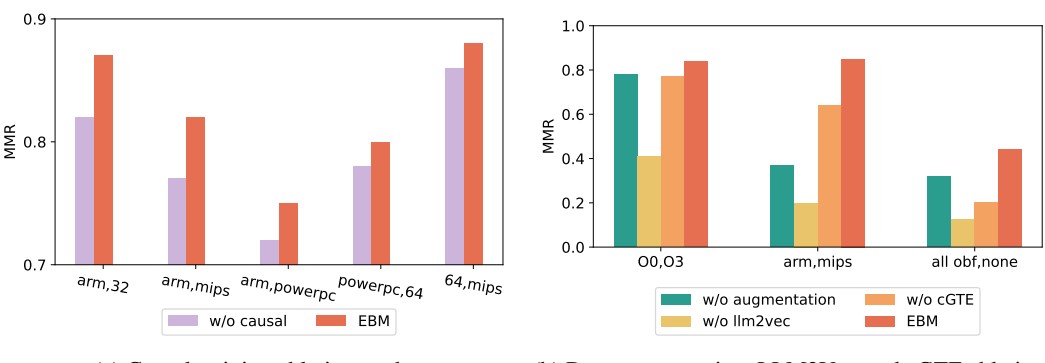

(a) Causal training ablation study       (b) Data augmentation, LLM2Vec, and cGTE ablation.

Figure 3: (3a) shows the MRR for two versions of EBM, with and without causal training. Cross-architecture retrieval is particularly sensitive to causal training, as it enables better translation between different languages. In (3b), EBM significantly improves the MRR compared to ablated models. LLM2Vec contributes the most to the increase.

## 6.1  Assembly Code Data Augmentation

We provide an ablation study on the data augmentation process, which includes data cleaning and adding special tokens for translation and structure awareness. The details can be found in Section 3.1. In Figure 3b, we plot the MRR when data augmentation is removed. It has a major improvement in cross-architecture and cross-obfuscation retrieval. Since these tasks are considered more difficult than cross-optimization retrieval, enhancing the quality of input data with structural and language information is a key process. The assembly code also becomes less noisy after the tokenization, which is often a constraint for cross-obfuscation detection.

## 6.2  Causal Training

Causal uptraining is the first process to fine-tune a generic LLM. In our analysis, it has an evident impact on improving cross-architecture retrieval and less impact on other retrieval tasks. In Figure 3a, we plot the cross-architecture evaluation for the two versions of our model, with and without causal uptraining. The MRR difference varies from 2% to 6%, which is a significant improvement.

## 6.3  LLM2Vec and cGTE

LLM2Vec is our BERT-like uptraining that converts a pre-trained decoder model to an embedding model by enabling bidirectional attention. cGTE is our enhanced version of InfoNCE loss during

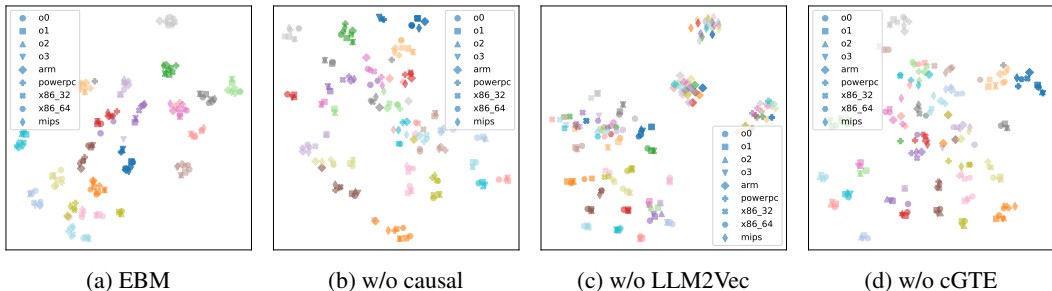

|          |          |          |          |
|:--------:|:--------:|:--------:|:--------:|
| (a) EBM  | (b) w/o causal | (c) w/o LLM2Vec | (d) w/o cGTE |

Figure 4: t-SNE visualization of embedding distributions. Each color corresponds to a binary function under different settings, represented by distinct shapes. That means, embeddings for the same function should cluster by color instead of shapes.

contrastive learning that introduces additional contrastive terms. Both are crucial to improving the retrieval performance, as shown in Figure 2. We show the difference in MRR for **[O0,O3]**, **[arm,mips]**, and **[all obf,none]**, as they represent cross-optimization, cross-architecture, and cross-obfuscation, respectively. Overall, LLM2Vec contributes the most to the performance gains for our training and can increase MRR by 2 to 4 times. cGTE is useful in complex tasks like cross-obfuscation, where introducing more negative contrastive pairs enables efficient training.

### 6.4  Embedding distributions

We provide visualizations for the embeddings of similar binary functions. The t-distributed Stochastic Neighbor Embedding (t-SNE) is used to reduce the embedding dimension to 2. In Figure 4, we plot 20 functions that are compiled using different optimization levels and architectures, represented in different shapes. Similar functions are grouped by colors, whereas the compilation setting is grouped by shapes. After dimensionality reduction, EBM produces tight groups that can be easily separated from the others. This illustrates that our final embeddings are effective and contain rich semantics.

## 7  Discussions and Limitations

LLMs have proven to be powerful for many learning tasks. To bridge such a technological gap in the binary software and security domain, we propose EBM, a multi-phase uptraining framework for binary code embedding and similarity detection. EBM is a flexible framework for any generic language or coder models, and is effective in outperforming all state-of-the-art benchmarks. Our data augmentation is carefully engineered to provide enhanced awareness of the language and structure information. The causal uptraining module leverages such information and pairs of data to translate different compilation settings to specifically enable cross-architecture learning phrased as a translation task. LLM2Vec performs a BERT-like task to learn the context and provide semantic-rich embeddings. Lastly, our custom cumulative GTE loss can efficiently capture more negative relationships during contrastive learning, significantly improving the semantics embedding in a resource-limited training environment. Our comprehensive evaluation shows that EBM outperforms all benchmarks in all tasks. The ablation study further indicates the effectiveness of all training processes.

**Limitations** Our limitations include the lack of evaluation on larger LLMs (due to our limited computational resources), relatively small training data compared to state-of-the-art coder models, and a lack of potential downstream tasks for binary code, such as malware and software vulnerability detection. We would like to lay a foundation for future work to apply our method or variants of it that address these limitations. We believe that such an approach has large potential in achieving the state-of-the-art going forward.

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

## A  More Related Work on Large Language Models

**LLM Architectures.** In the foundation of LLMs, transformer [Vaswani, 2017] is the architecture used to capture contextual relationships for long text sequences. The original transformer contains both encoder and decoder modules, where they work in tandem to encode input sequences into a fixed-length vector and output predictions for a new sequence. Such architecture can be used for machine translation, question answering, etc. Encoder models like BERT [Devlin, 2018] and its variants [Liu, 2019; Lan, 2019; Clark, 2020] perform pre-training using masked language modeling with a large amount of text data and generate contextual representations. BERT-style models use bi-directional information that conditions the entire sentence. The generated embeddings can be used for transfer learning or supervised fine-tuning in other downstream tasks. Decoder autoregressive (AR) models such as GPT [Radford, 2018] and Llama [Touvron *et al.*, 2023] perform the next token prediction based on previously seen tokens and generate new sequences.

**Foundation Coder Models** are trained on multiple programming languages and large codebases. They are capable of code generation, summarization, and translation. A list of modern open source coder models include Code Llama [Roziere *et al.*, 2023], codeT5 [Wang *et al.*, 2021], Qwen-coder [Hui *et al.*, 2024], GraphCodeBERT [Guo *et al.*, 2020], and many more. However, most of coder models are only trained using source code such as Python and Java. Assembly code has not attracted great attention from LLMs, thus some form of transfer learning is required to learn both the syntax and semantics of assembly code.

## B  Additional Results

This section contains additional results for cross-compiler (Table 7), cross-architectures (Table 9), and cross-obfuscation (Table 11). We also plot the t-SNE distributions for all LLM models in Figure 5.

| Models | MRR | Recall@1 |
|---|---|---|
| | clang,gcc | clang,gcc |
| SAFE | 0.200 | 0.063 |
| PalmTree | 0.423 | 0.178 |
| Asm2Vec | 0.523 | 0.328 |
| OrderMatters | 0.007 | 0.001 |
| GraphCodeBERT (125M) | 0.611 | 0.540 |
| CodeT5+ (110M) | 0.635 | 0.569 |
| Qwen2.5-Emb (1.5B) | 0.667 | 0.593 |
| Qwen2.5-Coder (1.5B) | 0.867 | 0.807 |
| EBM (0.5B) | **0.946** | **0.912** |

Table 7: Experimental results for cross-compiler between Clang and GCC. As a fairly easy task, Qwen2.5-Emb achieves 0.667 MRR with a zero-shot setting. EBM outperforms all the baselines by over 10%.

| Models | MRR | | | | | | |
|---|---|---|---|---|---|---|---|
| | Arm, PowerPC | Arm, x32 | Arm, MIPS | PowerPC, x32 | PowerPC, MIPS | x32, x64 | x32, MIPS |
| SAFE | 0.180 | 0.191 | 0.189 | 0.165 | 0.110 | 0.196 | 0.152 |
| PalmTree | 0.021 | 0.008 | 0.019 | 0.008 | 0.028 | 0.018 | 0.013 |
| Asm2Vec | 0.270 | 0.236 | 0.223 | 0.265 | 0.240 | 0.417 | 0.307 |
| OrderMatters | 0.008 | 0.008 | 0.007 | 0.008 | 0.006 | 0.007 | 0.008 |
| GraphCodeBERT (125M) | 0.094 | 0.075 | 0.064 | 0.256 | 0.215 | 0.495 | 0.239 |
| CodeT5+ (110M) | 0.057 | 0.043 | 0.048 | 0.287 | 0.259 | 0.461 | 0.259 |
| Qwen2.5-Emb (1.5B) | 0.014 | 0.030 | 0.032 | 0.142 | 0.095 | 0.409 | 0.079 |
| Qwen2.5-Coder (1.5B) | 0.269 | 0.319 | 0.374 | 0.490 | 0.480 | 0.757 | 0.522 |
| EBM (0.5B) | **0.718** | **0.818** | **0.770** | **0.836** | **0.803** | **0.906** | **0.883** |

Table 8: Additional MRR results for cross-architecture settings.

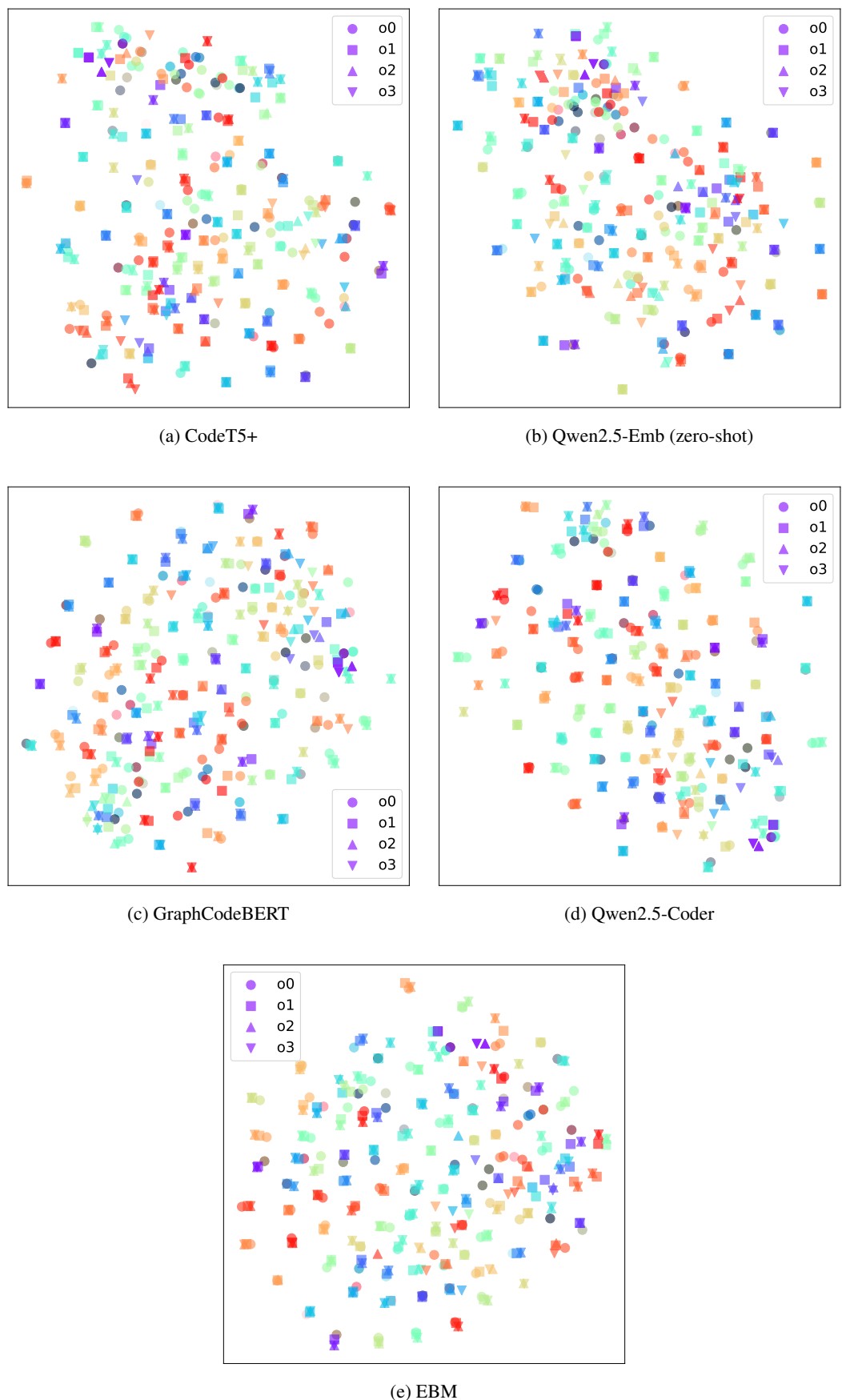

Figure 5: t-SNE visualization of embedding distributions of 100 functions generated by LLM models. Each color corresponds to a binary function under different optimization settings, represented by distinct shapes. Embeddings for the same function should cluster by color instead of shapes. Note that some colors are visually similar but represent different functions. It is evident that the embedding representations generated by EBM distribute more evenly in the space than in all others.

| Models | Recall@1 | | | | | | |
|---|---|---|---|---|---|---|---|
| | Arm, PowerPC | Arm, x32 | Arm, MIPS | PowerPC, x32 | PowerPC, MIPS | x32, x64 | x32, MIPS |
| SAFE | 0.063 | 0.000 | 0.063 | 0.000 | 0.000 | 0.063 | 0.000 |
| PalmTree | 0.001 | 0.001 | 0.001 | 0.001 | 0.001 | 0.007 | 0.000 |
| Asm2Vec | 0.098 | 0.080 | 0.072 | 0.085 | 0.080 | 0.231 | 0.114 |
| OrderMatters | 0.001 | 0.002 | 0.000 | 0.001 | 0.000 | 0.000 | 0.000 |
| GraphCodeBERT (125M) | 0.059 | 0.034 | 0.029 | 0.187 | 0.146 | 0.419 | 0.183 |
| CodeT5+ (110M) | 0.023 | 0.013 | 0.019 | 0.223 | 0.198 | 0.392 | 0.205 |
| Qwen2.5-Emb (1.5B) | 0.003 | 0.010 | 0.013 | 0.093 | 0.054 | 0.331 | 0.050 |
| Qwen2.5-Coder (1.5B) | 0.189 | 0.230 | 0.280 | 0.395 | 0.379 | 0.682 | 0.414 |
| EBM (0.5B) | **0.620** | **0.726** | **0.668** | **0.771** | **0.727** | **0.846** | **0.809** |

Table 9: Additional recall@1 results for cross-architecture settings.

| Models | MRR | | | | | | |
|---|---|---|---|---|---|---|---|
| | all, sub | all, fla | all, bcf | none, sub | none, fla | sub, bcf | fla, bcf |
| SAFE | 0.209 | 0.148 | 0.229 | 0.312 | 0.200 | 0.128 | 0.178 |
| PalmTree | 0.120 | 0.254 | 0.144 | 0.829 | 0.260 | 0.276 | 0.141 |
| Asm2Vec | 0.205 | 0.238 | 0.259 | 0.534 | 0.332 | 0.340 | 0.357 |
| OrderMatters | 0.008 | 0.008 | 0.007 | 0.007 | 0.008 | 0.008 | 0.008 |
| GraphCodeBERT (125M) | 0.239 | 0.331 | 0.290 | 0.932 | 0.464 | 0.621 | 0.447 |
| CodeT5+ (110M) | 0.174 | 0.324 | 0.263 | 0.873 | 0.378 | 0.585 | 0.465 |
| Qwen2.5-Emb (1.5B) | 0.277 | 0.337 | 0.355 | 0.900 | 0.466 | 0.618 | 0.463 |
| Qwen2.5-Coder (1.5B) | 0.374 | 0.417 | 0.422 | 0.953 | 0.583 | 0.687 | 0.540 |
| EBM (0.5B) | **0.522** | **0.704** | **0.614** | **0.989** | **0.815** | **0.817** | **0.790** |

Table 10: Additional MRR results for cross-obfuscation settings.

| Models | Recall@1 | | | | | | |
|---|---|---|---|---|---|---|---|
| | all, sub | all, fla | all, bcf | none, sub | none, fla | sub, bcf | fla, bcf |
| SAFE | 0.063 | 0.000 | 0.063 | 0.125 | 0.063 | 0.000 | 0.000 |
| PalmTree | 0.069 | 0.125 | 0.072 | 0.599 | 0.117 | 0.186 | 0.073 |
| Asm2Vec | 0.075 | 0.090 | 0.114 | 0.336 | 0.148 | 0.161 | 0.176 |
| OrderMatters | 0.001 | 0.001 | 0.001 | 0.000 | 0.002 | 0.001 | 0.002 |
| GraphCodeBERT (125M) | 0.186 | 0.268 | 0.230 | 0.901 | 0.384 | 0.540 | 0.375 |
| CodeT5+ (110M) | 0.127 | 0.251 | 0.196 | 0.839 | 0.301 | 0.511 | 0.374 |
| Qwen2.5-Emb (1.5B) | 0.207 | 0.262 | 0.284 | 0.867 | 0.375 | 0.540 | 0.380 |
| Qwen2.5-Coder (1.5B) | 0.292 | 0.340 | 0.340 | 0.921 | 0.504 | 0.603 | 0.454 |
| EBM (0.5B) | **0.436** | **0.624** | **0.523** | **0.972** | **0.751** | **0.751** | **0.717** |

Table 11: Additional recall@1 results for cross-obfuscation settings.

| Model | Average Inference Time per Batch (100 samples) in seconds | Average Training Time per Batch (4 samples) in it/s |
|---|---|---|
| CodeT5P(110M) | 0.007 | 13.90 |
| GraphCodeBERT(125M) | 0.005 | 17.78 |
| Qwen2.5(1.5B) | 2.000 | 2.78 |
| CodeGemma2B | 2.540 | 2.12 |
| EBM(0.5B) | 0.014 | 3.32 |

Table 12: Model Computational Performance Comparison

| Source | Dest | Accuracy | | Mean ± STD of Similar pairs | | Mean ± STD of Non-similar pairs | |
|--------|------|----------|----|-----------------------------|----|--------------------------------|----|
| | | CodeGemma | EBM | CodeGemma | EBM | CodeGemma | EBM |
| obf_all | obf_none | 0.207 | 0.103 | 0.615 ± 0.232 | 0.580 ± 0.216 | 0.141 ± 0.189 | 0.244 ± 0.156 |
| obf_all | obf_sub | 0.179 | 0.097 | 0.607 ± 0.235 | 0.592 ± 0.206 | 0.149 ± 0.189 | 0.254 ± 0.158 |
| obf_all | obf_fla | 0.057 | 0.085 | 0.719 ± 0.185 | 0.765 ± 0.145 | 0.221 ± 0.192 | 0.386 ± 0.172 |
| obf_all | obf_bcf | 0.111 | 0.026 | 0.651 ± 0.227 | 0.690 ± 0.166 | 0.172 ± 0.189 | 0.347 ± 0.163 |
| obf_none | obf_sub | 0.838 | 0.890 | 0.943 ± 0.059 | 0.972 ± 0.065 | 0.181 ± 0.222 | 0.210 ± 0.179 |
| obf_none | obf_fla | 0.169 | 0.341 | 0.763 ± 0.167 | 0.746 ± 0.161 | 0.166 ± 0.206 | 0.230 ± 0.167 |
| obf_none | obf_bcf | 0.354 | 0.320 | 0.843 ± 0.149 | 0.774 ± 0.180 | 0.180 ± 0.218 | 0.233 ± 0.172 |
| obf_sub | obf_fla | 0.120 | 0.278 | 0.740 ± 0.172 | 0.734 ± 0.161 | 0.161 ± 0.202 | 0.235 ± 0.165 |
| obf_sub | obf_bcf | 0.231 | 0.265 | 0.810 ± 0.159 | 0.762 ± 0.173 | 0.169 ± 0.210 | 0.232 ± 0.171 |
| obf_fla | obf_bcf | 0.176 | 0.273 | 0.766 ± 0.167 | 0.735 ± 0.141 | 0.184 ± 0.195 | 0.291 ± 0.162 |
| clang | gcc | 0.529 | 0.849 | 0.911 ± 0.105 | 0.896 ± 0.097 | 0.200 ± 0.257 | 0.103 ± 0.164 |
| arm | powerpc | 0.014 | 0.478 | 0.727 ± 0.173 | 0.765 ± 0.168 | 0.278 ± 0.270 | 0.111 ± 0.160 |
| arm | x86_32 | 0.014 | 0.624 | 0.732 ± 0.149 | 0.801 ± 0.137 | 0.290 ± 0.266 | 0.108 ± 0.166 |
| arm | x86_64 | 0.011 | 0.526 | 0.724 ± 0.173 | 0.776 ± 0.163 | 0.273 ± 0.267 | 0.102 ± 0.161 |
| arm | mips | 0.034 | 0.587 | 0.757 ± 0.149 | 0.805 ± 0.148 | 0.283 ± 0.266 | 0.094 ± 0.159 |
| powerpc | x86_32 | 0.111 | 0.529 | 0.820 ± 0.157 | 0.810 ± 0.157 | 0.307 ± 0.267 | 0.123 ± 0.173 |
| powerpc | x86_64 | 0.102 | 0.424 | 0.818 ± 0.171 | 0.797 ± 0.195 | 0.279 ± 0.270 | 0.119 ± 0.169 |
| powerpc | mips | 0.104 | 0.474 | 0.803 ± 0.165 | 0.792 ± 0.181 | 0.302 ± 0.261 | 0.114 ± 0.165 |
| x86_32 | x86_64 | 0.276 | 0.712 | 0.902 ± 0.146 | 0.877 ± 0.148 | 0.318 ± 0.262 | 0.121 ± 0.179 |
| x86_32 | mips | 0.136 | 0.663 | 0.797 ± 0.144 | 0.826 ± 0.140 | 0.280 ± 0.263 | 0.103 ± 0.166 |
| x86_64 | mips | 0.124 | 0.624 | 0.800 ± 0.145 | 0.810 ± 0.143 | 0.286 ± 0.256 | 0.103 ± 0.165 |

Table 13: Additional results for Model Accuracy for non-match and Similarity Metrics Comparison

