# OpenReview forum: "Transforming Generic Coder LLMs to Effective Binary Code Embedding Models for Similarity Detection"
_NeurIPS.cc/2025/Conference — NeurIPS 2025 poster_

### Official Review · Reviewer_3vSh · 2025-06-22

**Clarity:** 3
**Significance:** 4
**Originality:** 3
**Rating:** 5
**Confidence:** 4

**Summary:**

Vulnerability detection in binary code is a very challenging and the generic large language models are not yet mature enough to reason about similar code. This work proposes uplearning the pretrained LLM to identify the binary code similarity and has shown effectiveness against zero-shot LLMS and the state-of-the-art binary code similarity detection methods.

**Questions:**

The equations 1 & 2 are not explained

**Ethical Concerns:**

["NO or VERY MINOR ethics concerns only"]

**Limitations:**

Limitations are included under the conclusion. Limitations include the quantity of the training data set compared to previous work, and the lack of potential downstream tasks, etc. However, this work can be good initial work achieve these limitations.

**Paper Formatting Concerns:**

No noticeable formatting issues

**Quality:**

4

**Strengths And Weaknesses:**

Strengths :
The causal relationship is vital in reasoning the binary code, specifically in understanding the flow. This work is uptrained the LLMS with the causal training which can improve the LLMs capabilities against binary code.

Weaknesses:
I do not see any specific weakness in this work

---

> ### Author Response · Authors · 2025-08-08
>
> Thank you very much for the review. We are aware of the downstream tasks and can add more discussions in the limitation section.

---

### Official Review · Reviewer_W13r · 2025-06-29

**Clarity:** 3
**Significance:** 2
**Originality:** 3
**Rating:** 4
**Confidence:** 4

**Summary:**

This paper explores using LLMs for binary code similarity detection. It demonstrate that pretrained LLMs can detect similar binary code even in a zero-shot setting, and proposes supervised fine-tuning methods, including data augmentation, translation-style learning, LLM2Vec, and cumulative GTE loss, that significantly outperform zero-shot LLMs and state-of-the-art approaches. The approach enhances robustness across different optimizations, architectures, and obfuscations, transforming generic LLMs into effective binary similarity experts.

**Questions:**

1. How does the backbone model (Qwen2.5-Coder-0.5B) perform without training? (This question is concerned to show the improvement from the approach)

2. How does the approach perform when applied to different backbone models with different scales of parameters? (This question is concerned on the generality of the approach as currently the evaluation is restricted on one small backbone model).

3. What are the downstream application scenarios of the BCSD techniques in practice? (This question is related to the importance and motivation of the studied program, also, it could help justify why the BCSD can be formulated as a retrieval problem).

4. How is the retrieval pool constructed? Why use the size 1000 by default?

**Ethical Concerns:**

["NO or VERY MINOR ethics concerns only"]

**Final Justification:**

The rebuttal has partially addressed my concerns.

The experiments are still limited to Qwen models as backbone models, so I have concerns regarding generalization. Moreover, the parameter setting is arbitrary, and the authors acknowledge it.

**Limitations:**

Yes.

**Paper Formatting Concerns:**

The ids of sub-sections start from 0 not 1 (e.g., 6.0.1).

**Quality:**

2

**Strengths And Weaknesses:**

## Strengths
+ The evaluation includes diverse scenarios from multiple perspectives (e.g., cross-optimization, cross-architecture, cross-compiler, cross-obfuscation).

## Weaknesses
- The generality of the approach is not well justified empirically given only one small backbone model is involved.
- The downstream application tasks of the BCSD techniques are missing (in terms of both the discussion and empirical evaluation). It hurts the significance of interpreting the metric improvements of the proposed approach in practice.

---

> ### Author Response · Authors · 2025-08-01
> **Addressing weaknesses and questions**
>
> Dear reviewer,
>
> We appreciate and value your feedback on our work and would like to provide some comments.
>
> Weakness 1: Generality of the approach
>
> Our work provides a general framework that can be applied to autoregressive language models. We have conducted additional experiments on the Qwen3-1.7B model, which belongs to the new Qwen3 series, which are shown to outperform coding and math tasks than many state-of-the-art LLMs like Deepseek and LLAMA. Due to time and computational resource constraints, we trained it with fewer epochs and applied LoRA in the final contrastive learning stage. We illustrate average MRR on the evaluation settings compared to our original results trained on Qwen2-0.6B:
>
> Qwen2: Cross-optimization: 0.916 Cross-architecture: 0.811 Cross-obfuscation: 0.710
>
> Qwen3: Cross-optimization: 0.910 Cross-architecture: 0.825 Cross-obfuscation: 0.682
>
> Given the above results, we believe that our training framework can be applied to other models as well. Moreover, we hypothesize that the performance can even be scaled with model parameter size, as significantly less training on the 1.7B model still leads to comparable results.
>
> Please understand that due to limited computational resources, we are unable to train larger models with tens of billions of parameters. Moreover, modern embedding models tend to be sufficient with a small parameter count. For example, the largest Qwen3 embedding model has only 8B parameters.
>
> Weakness 2: Discussion of downstream tasks
>
> The downstream tasks for BCSD include clone/code search, vulnerability discovery, and malware detection. The most common use case is to query a given binary function on a known vulnerability/malware database to retrieve similar functions, potentially identifying exploitable/malicious code. We do not evaluate the empirical results for such tasks because these are usually studied as separate research/engineering problems from BCSD. In this work, we focus on a controlled experiment setting for robust binary code embedding given various compile settings. We are happy to include a discussion of the application of BCSD in the paper in the next stage.
>
> Question 1: Performance of the backbone model without training
>
> Autoregressive models should be used as generative models rather than embedding models. We did not use it as a baseline as we deemed it unusable without fine-tuning. We included the Qwen2.5-Emb model, which is an embedding model, as one of the baselines in our experiment. Regardless, we can show the average MRR for all three settings for Qwen2.5-Coder-0.5B:
>
> Cross-optimization: 0.404
> Cross-obfuscation: 0.177
> Cross-architecture: 0.047
>
> Question 2: Please refer to weakness 1
>
> Question 3: Please refer to weakness 2
>
> Question 4: Construction of the retrieval pool
>
> The retrieval pool is constructed by randomly sampling 1000 function names for each task (i.e. O0-O3, Arm-x64) and storing the keys for reproducibility. Choosing 1000 is arbitrary. It represents somewhat of a real-world use case and also does not exceed the total number of functions in the evaluation libraries. As long as the pool contains the same functions to retrieve from and no variability is introduced between different models and runs, we believe this is a good representation of the evaluation.
>
> Please let us know if you have further questions. We look forward to hearing your feedback.

---

> > ### Comment · Reviewer_W13r · 2025-08-04
> >
> > Thanks for the rebuttal. The provided additional results have partially addressed my concerns, so I have raised the score to 4.

---

### Official Review · Reviewer_CBZw · 2025-06-30

**Clarity:** 3
**Significance:** 2
**Originality:** 2
**Rating:** 4
**Confidence:** 4

**Summary:**

This paper considers the problem of training embedding models for detecting if a pair of code binaries are similar, i.e., generated by compiling the same source file but with different compiler options or architecture backends. For this purpose, the paper presents a method for fine-tune existing code models using a combination of causal language modeling, masked language modeling, and contrastive learning objectives. A model (Qwen2.5-Coder-0.5B) fine-tuned with these objectives is compared with various baselines on similar binary code retrieval task. Experiments show that the fine-tuned model outperforms various baselines.

**Questions:**

1. Why not consider pairs of binaries that are compiled from different, but semantically equivalent source codes?

2. Does the method require metadata about the binaries such as the hardware architecture, the compiler optimization levels, etc? If so, can this information be extracted directly from binary files? If not, how will the method be used in practice?

3. Equation 6 is not clear. What do you mean by "multiple distributed models" and "accumulated inputs"?

4. Does the method require disassembling the binary first? Section 3.1 seems to suggest this. It would be good to clarify this.

5. Which dataset is used for results in Fig 3?

**Ethical Concerns:**

["NO or VERY MINOR ethics concerns only"]

**Final Justification:**

I thank the authors for their response which helped clarify some conceptual issues. I will maintain my score.

**Limitations:**

The paper has a brief mention of limitations in the conclusion. Perhaps, this could be expanded. One limitation that could be discussed is whether the method is applicable for two binaries that are compiled from different but semantically equivalent source files.

**Quality:**

3

**Strengths And Weaknesses:**

### Strengths
1. The empirical results are very promising.

2. The proposed training methods are fairly straightforward, which is a good thing! It seems like the methods should be easily applicable to other baseline models as well.

### Weaknesses
1. It would have been nice if the experiments included results from fine-tuning another model. At the moment, one cannot say for sure if the presented method will show improvements on other models as well.

2. The experiments only consider similarity between pairs of binaries that are compiled from the same source code. I am very curious what the results look like for a pair of binaries that are compiled from different, but semantically equivalent source codes.

3. If I understand correctly, during inference, the method requires access to metadata about the binaries such as the hardware architecture, the compiler optimization levels, etc. Can such information be extracted from binary files? If not, the method may not practically applicable.

---

> ### Author Response · Authors · 2025-08-01
> **Addressing the weaknesses and questions**
>
> Dear reviewer,
>
> We appreciate and value your feedback on our work and would like to provide some comments.
>
> Weakness 1: Inclusion of other models
>
> Our work provides a general framework that can be applied to autoregressive language models. We have conducted additional experiments on the Qwen3-1.7B model, which belongs to the new Qwen3 series, which are shown to outperform coding and math tasks than many state-of-the-art LLMs like Deepseek and LLAMA. Due to time and computational resource constraints, we trained it with fewer epochs and applied LoRA in the final contrastive learning stage. We illustrate average MRR on the evaluation settings compared to our original results trained on Qwen2-0.6B:
>
> Qwen2:
> Cross-optimization: 0.916
> Cross-architecture: 0.811
> Cross-obfuscation: 0.710
>
> Qwen3:
> Cross-optimization: 0.910
> Cross-architecture: 0.825
> Cross-obfuscation: 0.682
>
> Given the above results, we believe that our training framework can be applied to other models as well. Moreover, we hypothesize that the performance can even be scaled with model parameter size, as significantly less training on the 1.7B model still leads to comparable results.
>
> Weakness 2: performance on different but semantically equivalent source codes
>
> We would like to clarify that the obfuscation evaluation serves exactly this purpose. The idea of obfuscation is to syntactically and structurally modify the code without any semantic changes. If we reverse engineer the different obfuscated binary codes back to source code, they would have different but semantically equivalent codes. We want to admit that obfuscation remains the biggest challenge for code matching and should be researched with more depth in the future.
>
> Weakness 3: Inference requires access to metadata
>
> It was not made clear in our paper that the introduction of language tokens is entirely for training the model to perform translation-style training. We are hoping to add a statement to clarify that. We are aware that during inference, it is impossible to retrieve the exact compiler settings. In our experiments, language tokens are absent, and only the BLK tokens are included for structural awareness, and such tokens are available when disassembling the object file to assembly code.
>
> Question 1: Please refer to weakness 2
>
> Question 2: Please refer to weakness 3
>
> Question 3: Clarification of equation 6
>
> We use parallel training to distribute batches across multiple GPUs. Since cGTE loss directly benefits from larger batch sizes, we collect all batches across GPUs first before calculating the loss. n represents the number of GPUs in this equation. We will modify the description of this equation to make it clearer.
>
> Question 4: Requirement of disassembly
>
> Yes, our method relies on access to the assembly code, as most other binary matching models do. Object or binary code alone does not provide meaningful information that can be encoded as the input to language models. Therefore, a higher level of representation is needed. We will clarify this in the final version.
>
> Question 5: Dataset for Figure 3
>
> The dataset used is the same test set that is evaluated in the experiment. However, we did notice that the reported MRR for EBM in this figure belongs to a slightly earlier version since the ablation study was completed then, and the performance is slightly worse in some cases than shown in our tables. Although the overall conclusion remains the same, we will make sure to update it later.
>
> Please let us know if you have further questions. We look forward to hearing your feedback.

---

> > ### Comment · Reviewer_CBZw · 2025-08-03
> >
> > Thank you for the response! Regarding weakness 2, perhaps my comment was not very clear, so let me rephrase. Suppose we have a program *S* in source code. The method in the paper is used to detect if binaries *B1* and *B2* that are generated from compiling *S* under different compiler options / architecture backends are similar. However, consider the situation where we have another source code program *S'* that is semantically equivalent to *S*. If *S* and *S'* are compiled with the same compiler options and architecture backends, would the presented method be able to detect that the two resulting binaries are similar?

---

> > > ### Author Response · Authors · 2025-08-04
> > >
> > > Thank you for the response. We understood your comment regarding weakness 2. The obfuscation techniques are applied to the code during build time. There are generally multiple ways to obfuscate code: pre-compile, during-compile, and post-compile. You suggested the evaluation on pre-compile (i.e. source code) obfuscation, where we evaluated during-compile obfuscation. Although we cannot confidently state that their effects are identical, they should be very similar when we examine the assembly code, i.e. more basic blocks, more loops in the CFG, etc. Due to time constraints, we are not able to fully evaluate the different obfuscation applications, but we can add these statements and limitations in our conclusion.

---

### Official Review · Reviewer_NsTz · 2025-07-03

**Clarity:** 3
**Significance:** 3
**Originality:** 3
**Rating:** 4
**Confidence:** 3

**Summary:**

This paper proposes a training strategy to transform an LLM into an assembly code embedding model. It flattens the code and adds structural "language" tokens for improved syntax and semantic representation. It treats function pairs like translation tasks to learn cross-variant semantics. It leverages LLM2Vec to enable bidirectional attention in causal models for better embeddings. It also utilizes cumulative GTE loss that enhances training with limited resources. Experiments are conducted on two datasets, and results show that it outperforms baselines.

**Questions:**

Please refer to the weaknesses.

**Ethical Concerns:**

["NO or VERY MINOR ethics concerns only"]

**Final Justification:**

The rebuttal addressed some of my concerns and I will keep my score.

**Limitations:**

It may not work when the binary code cannot be disassembled correctly.

**Quality:**

3

**Strengths And Weaknesses:**

Strengths:
1. Overall, it's well-written and easy to follow.
2. The evaluation considers multiple datasets and baselines, and also includes ablation studies on different components.
3. The results show that the proposed method outperforms baselines.

Weaknesses:
1. It would be better to include the time cost. Because this paper tries to explore an efficient fine-tuning method, it's suggested to show that the proposed method is more efficient than existing methods.
2. It would be better to evaluate some other cases.

    2.1 If the pool doesn't contain "exactly one similar function", that is, "no similar function" or "multiple similar functions", what will be the precision and recall?

    2.2 If the user doesn't know the language information of the binary code, or wrong language information is provided, what will happen?

    2.3 When the training configuration is different from the test one, how will the performance change? For example, if the model is only trained on o0, o3, can it work for o1, o3?

3. It would be better to introduce and explain the symbols in each equation. For example, $q_i, k_i, k_j, \tau, \eta, (n)$, etc.

---

> ### Author Response · Authors · 2025-08-01
>
> Dear reviewer,
>
> We appreciate and value your feedback on our work and would like to provide some comments.
>
> **Weaknesses 1**: We have provided the following table for the training time (larger is better) and inference time (smaller is better)
>
> | Model | Average Inference Time per Batch (100 samples) in seconds | Average Training Time per Batch (4 samples) in it/s |
> |----|------|--|
> | EBM(0.5B) | 0.014  |  3.32 it/s|
> | CodeGemma2B| 2.540  |  2.12 it/s |
> | CodeT5P(110M)| 0.007 | 13.90 it/s  |
> | GraphCodeBERT(125M) | 0.005 | 17.78 it/s  |
> | Qwen2.5(1.5B)       | 2.000 |  2.78 it/s  |
>
> **Weaknesses 2.1** We have rerun some baselines for "no similar function" in the pool by setting a threshold by *MEAN-STD* of the similar scores of the training pairs. The accuracy is calculated by (# of queries with no matching pair  / total number of test queries). EBM still shows decent performance and outperforms all the baselines.
>
> | source | dest| Accuracy (CodeGemma) | Accuracy (EBM) | Mean ± STD of Similar pairs (CodeGemma) | Mean ± STD of Similar pairs (EBM) | Mean ± STD of Non-similar pairs (CodeGemma) | Mean ± STD of Non-similar pairs (EBM) |
> |------------|-----------|----------------------|----------------|-----------|-------------|-----------|-------------|
> | o0 | o1 | 0.472 | 0.719 | 0.904 ± 0.136 | 0.910 ± 0.106| 0.207 ± 0.220 | 0.180 ± 0.180 |
> | o0 | o2| 0.387 | 0.676| 0.868 ± 0.155| 0.879 ± 0.120| 0.208 ± 0.212| 0.178 ± 0.172|
> | o0 | o3| 0.236 | 0.458| 0.830 ± 0.195| 0.833 ± 0.173| 0.210 ± 0.213| 0.184 ± 0.174|
> | o1 | o2| 0.675 | 0.831| 0.937 ± 0.092| 0.952 ± 0.076| 0.176 ± 0.213| 0.164 ± 0.173|
> | o1 | o3| 0.438 | 0.675| 0.899 ± 0.151| 0.916 ± 0.134| 0.168 ± 0.221| 0.170 ± 0.176|
> | o2 | o3| 0.618 | 0.793| 0.955 ± 0.139| 0.965 ± 0.133| 0.183 ± 0.217| 0.174 ± 0.173|
> | obf_all| obf_none| 0.207 | 0.103| 0.615 ± 0.232| 0.580 ± 0.216| 0.141 ± 0.189| 0.244 ± 0.156|
> | obf_all| obf_sub | 0.179 | 0.097| 0.607 ± 0.235| 0.592 ± 0.206| 0.149 ± 0.189| 0.254 ± 0.158|
> | obf_all| obf_fla | 0.057 | 0.085| 0.719 ± 0.185| 0.765 ± 0.145| 0.221 ± 0.192| 0.386 ± 0.172|
> | obf_all| obf_bcf | 0.111 | 0.026| 0.651 ± 0.227| 0.690 ± 0.166| 0.172 ± 0.189| 0.347 ± 0.163|
> | obf_none | obf_sub | 0.838 | 0.89 | 0.943 ± 0.059| 0.972 ± 0.065| 0.181 ± 0.222| 0.210 ± 0.179|
> | obf_none | obf_fla | 0.169 | 0.341| 0.763 ± 0.167| 0.746 ± 0.161| 0.166 ± 0.206| 0.230 ± 0.167|
> | obf_none | obf_bcf | 0.354 | 0.32 | 0.843 ± 0.149| 0.774 ± 0.180| 0.180 ± 0.218| 0.233 ± 0.172|
> | obf_sub| obf_fla | 0.12| 0.278| 0.740 ± 0.172| 0.734 ± 0.161| 0.161 ± 0.202| 0.235 ± 0.165|
> | obf_sub| obf_bcf | 0.231 | 0.265| 0.810 ± 0.159| 0.762 ± 0.173| 0.169 ± 0.210| 0.232 ± 0.171|
> | obf_fla| obf_bcf | 0.176 | 0.273| 0.766 ± 0.167| 0.735 ± 0.141| 0.184 ± 0.195| 0.291 ± 0.162|
> | clang| gcc | 0.529 | 0.849| 0.911 ± 0.105| 0.896 ± 0.097| 0.200 ± 0.257| 0.103 ± 0.164|
> | arm| powerpc | 0.014 | 0.478| 0.727 ± 0.173| 0.765 ± 0.168| 0.278 ± 0.270| 0.111 ± 0.160|
> | arm| x86_32| 0.014 | 0.624| 0.732 ± 0.149| 0.801 ± 0.137| 0.290 ± 0.266| 0.108 ± 0.166|
> | arm| x86_64| 0.011 | 0.526| 0.724 ± 0.173| 0.776 ± 0.163| 0.273 ± 0.267| 0.102 ± 0.161|
> | arm| mips| 0.034 | 0.587| 0.757 ± 0.149| 0.805 ± 0.148| 0.283 ± 0.266| 0.094 ± 0.159|
> | powerpc| x86_32| 0.111 | 0.529| 0.820 ± 0.157| 0.810 ± 0.157| 0.307 ± 0.267| 0.123 ± 0.173|
> | powerpc| x86_64| 0.102 | 0.424| 0.818 ± 0.171| 0.797 ± 0.195| 0.279 ± 0.270| 0.119 ± 0.169|
> | powerpc| mips| 0.104 | 0.474| 0.803 ± 0.165| 0.792 ± 0.181| 0.302 ± 0.261| 0.114 ± 0.165|
> | x86_32 | x86_64| 0.276 | 0.712| 0.902 ± 0.146| 0.877 ± 0.148| 0.318 ± 0.262| 0.121 ± 0.179 |
> | x86_32 | mips | 0.136 | 0.663| 0.797 ± 0.144| 0.826 ± 0.140| 0.280 ± 0.263| 0.103 ± 0.166 |
> | x86_64 | mips| 0.124| 0.624| 0.800 ± 0.145| 0.810 ± 0.143| 0.286 ± 0.256| 0.103 ± 0.165|
>
> **Weaknesses 2.1** Regarding "multiple similar functions", we will not put same functions in the pool. During inference, we will only pick top-1 as the detected one.
>
> **Weaknesses 2.2** On line 228, it is mentioned that the language information is not used during inference. In addition, if you ask if we need to know the source binary settings in order to search the pool, since we have tested all the combinations of binary settings, it is reasonable to claim we can detect the similar pairs in any destination pool without knowing the source settings.
>
> **Weaknesses 2.3** We train all the combinations during training. However, neglecting "o1, o3" during training will definitely underperform "o1, o3" during inference. But the point is that if we train some pairs of "o1, o3", the model is general enough for new unseen pairs of "o1,o3".
>
> **Weaknesses 3** Thanks for pointing out. $q$ means query, $k$ means key. $\tau$ represents the temperature hyperparameter configuration for cGTE (we set to 0.05) $\eta$ is the learning rate.

---

> > ### Comment · Reviewer_NsTz · 2025-08-04
> >
> > Thanks for the response, and I will maintain my score.

---

### Official Review · Reviewer_va3T · 2025-07-04

**Clarity:** 1
**Significance:** 1
**Originality:** 1
**Rating:** 1
**Confidence:** 4

**Summary:**

This paper studies the task of binary executable code similarity detection with help of LLMs. They propose finetuning an LLM model on assembly data and show that this finetuned model outperforms similar models evaluated in zero-shot setting.

**Questions:**

- "the same source code can be compiled into many formats of binary code using different compiler settings and environments, including optimization levels, machine architectures, compiler software, and obfuscation techniques. " - The same is true for source code similarity detection, the same functionality can be written in many ways in source code. While this is a limitation and potential challenge, further motivation is needed to understand why the LLMs that work for source

- On line 193: "As our model is based upon Qwen2.5-Coder-0.5B, the Qwen2.5-Coder-1.5B baseline is a good candidate for ablation study." - not clear why you suggest it is an especially good candidate compared to any other coding model? Why that size of the model?

- Authors mention they finetune all models with their contrastive loss, except Qwen-Emb, which is supposed to illustrate the difference between code & non-code models, but that makes no sense to me. Why also not finetune Qwen-Emb and show that code models perform better compared to non-code ones, even when finetuned? Or perhaps authors meant something else, but it wasn't clear to me.

**Ethical Concerns:**

["NO or VERY MINOR ethics concerns only"]

**Limitations:**

yes

**Paper Formatting Concerns:**

The paper is poorly formatted (e.g. references to prior work inserted in the text makes it unreadable, odd choice of using subsubsections without subsections, etc)

**Quality:**

1

**Strengths And Weaknesses:**

**Weaknesses**

- The paper is poorly written; the motivations behind decisions made in the study and proposed novelty are not spelled out; some of the contributions mentioned, such as "comparing baselines", are not sufficient to be considered a contribution in my opinion. There are multiple ambiguous pronoun references which affects the clarity of the paper, e.g. in Conclusion; acronyms are used without prior introduction, e.g. what is AR on line 139? GTE on line 173?

- Key details are missing, e.g. on line 126 the authors say: "we add several additional tokens to indicate the “language" information of each binary function, which corresponds to the optimization level, compiler, obfuscation, and architecture. Adding such information has proven to be effective, especially for cross-architecture detection." - without specifying what tokens, how and where they are added, at what stage, etc, this information is not as useful for the reader as it could be otherwise. In general, the paper does not seem to be written with AI professional readers in mind, e.g. the authors go into details explaining how autoregressive models learn, but gloss over what their used obfuscation methods are.

- The authors suggest that their proposed solution based on continual training and finetuning of an LLM model is cheap and effective, which is questionable considering approaches like parameter-efficient FT or inference-time demonstrations with larger models are not considered at all.

- The choice of the baseline models and how each one is trained is unclear to me. The authors mix embedding models, such as CodeT5+ and GraphCodeBERT, with generative models, such as Qwen2.5-Coder1.5 for a task of ranking pairs. How are they compared is not clear. This weakens the results because it is not clear what comparisons are being drawn and which of the proposed modifications/design choices are responsible for performance report.

- The prior work section is incomplete when it comes to recent developments, and omits most recent works and models, instead referring back and comparing to 5+ years old models, such as GraphCodeBERT and CodeT5+, which are not state of the art, and weren't intended to be used with assembly either, so not clear why those were chosen over other more modern & stronger code models. To make the paper and it's findings valuable it would need to draw comparisons to more recent models such as DeepSeek models, SantaCoder, WizardCoder, CodeGemma, CodeLlama, etc.. The authors use Qwen-Coder as their base model, which is a strong, state-of-the-art model.

- The motivation behind performing continual training on a generative model and using it as an embedding model is not clear, and most of the design choices are not intuitive, and neither are they explained or supported, which limits the usefullness of the paper since the takeaway messages are not clear. It is also not at all surprising that finetuning an LLM may outperform other, similarly-sized LLMs that are evaluated in zero-shot manner, so the whole evaluation setup needs an overhaul to provide new knowledge for the readers.


**Strengths**

The authors use a comprehensive dataset of cross-architecture, cross-optimization, cross-compiler, cross-obfuscation binaries for their experimentation.

---

> ### Author Response · Authors · 2025-08-05
>
> Sorry about the confusion we've made. We hope to address your concerns. Please respect our efforts and we appreciate if you can rate our work again.
>
> **W1** Since we have done a lot of experiments and each one require a decent amount of time. We want to treat it as one of our contributions. AR means auto-regressive and GTE means general text embedding. We will update them if possible.
>
> **W2** On line 112, we explain three special tokens we used to replace memory address, string, and bytes. I agree that we can add evaluations about the how these tokens can impact the model weights. But we do add results in Fig3b to indicate the impacts on the final performance.
>
> **W3** Please note that we also use parameter-efficient FT like lora for large models. Our approach is efficient in terms of smaller models with better performance. In addition, generative models are not applicable to embedding tasks, where embedding models generate one vector only. As a result, inference time scaling is not considered.  In addition, we have provided the following table for the training time (larger is better) and inference time (smaller is better)
>
> | Model | Average Inference Time per Batch (100 samples) in seconds | Average Training Time per Batch (4 samples) in it/s |
> |----|------|--|
> | EBM(0.5B) | 0.014  |  3.32 it/s|
> | CodeGemma2B| 2.540  |  2.12 it/s |
> | CodeT5P(110M)| 0.007 | 13.90 it/s  |
> | GraphCodeBERT(125M) | 0.005 | 17.78 it/s  |
> | Qwen2.5(1.5B)       | 2.000 |  2.78 it/s  |
>
> **W4** As mentioned in previous comment, the whole point of this paper is to provide an approach to turn generative models into embedding models. That is why we mix embedding baseline and generative models. Also,  on line 191, we mentioned all the models are ft-ed (except Qwen emb) by contrastive loss before comparison.
>
> **W5** Thanks for the comment. We've provided additional results for latest models: Codegemma 2B
>
> Cross Optimization (Align to table 1)
> CodeGemma (2B)  & 0.763 & 0.888 & 0.833 & 0.866 & 0.931 & 0.856 & 0.696 & 0.840 & 0.778 & 0.821 & 0.905 & 0.808
>
> Cross Architecture (Align to table 2)
> CodeGemma (2B)       & 0.293 & 0.581 & 0.548 & 0.474 & 0.208 & 0.479 & 0.432 & 0.373
>
> Cross Obfuscation (Align to table 3)
> CodeGemma (2B) & 0.454 & 0.796 & 0.571 & 0.607 & 0.356 & 0.735 & 0.473 & 0.521
>
> Indeed larger and latest models generate better results. But EBM still outperforms Codegemma 2B.
>
> **W6** On line 190, we mentioned only Qwen-Emb is used as a zero-shot manner. On the same line, we explained the reason is that we want to demonstrate to readers about the performance of applying code embedding models to binary code, such that it is not a trivial task to embed binary code. Also, Qwen-Emb 1.5 is not a similar-sized model with EBM 0.5B.
>
> Short summary: It seems that the choice of baslines is a great concern. If we compare the baselines in the following way, will it be clearer to you?
>
> the performance between a generative model trained to an embedding model with the contrasive loss vs the same generative model by EBM. And we can make a couple of comparisons in such way.
>
> **Q1** In this paper, we limit the scope to binary code similarity detection (BCSD) where source code is not applicable during real life scenarios.
>
> **Q2** Since EMB is uptrained from Qwen2.5-0.5B, the fact that EBM outperforms Qwen2.5-1.5B (with just contrastive loss) validates the our approach. In another word, it's a good baseline candidate because they have same architecture, same training data, and 1.5B model has even more parameters.
>
> **Q3**
> We indeed fine-tuned Qwen2.5-coder-1.5 to an embeding model with the same training data with EBM.
>
> Please kindly reconsider your rating since our experiment results show a clear boost to the final performance. And we believe our work can benefit the BCSD community. Thanks

---

> > ### Comment · Reviewer_va3T · 2025-08-05
> > **Acknowledgement of rebuttal past deadline.**
> >
> > Thank you for responding to my review. However, per NeurIPS policy, your rebuttal to my review cannot be considered:
> > >We regret that any rebuttals that were missing by the rebuttal deadline and were posted in their entirety after Jul 30 (e.g., by using the comment button) are to be ignored. Unfortunately, it is unfair on the vast amount of authors who “did the right thing” to count such late rebuttals.

---

> > > ### Author Response · Authors · 2025-08-05
> > >
> > > Thanks for responding. You surely remain the right to do so. However, could you consider this as a discussion (as we can still discuss during discussion period)?
> > >
> > > Could you please carefully read our paper again and leave some comments to each point? We also have the right to ask for additional evidence why we were given “1” point for each criterion, i.e.
> > > - writing quality given the reason for two unexplained acronyms.
> > > - why is our work not significant
> > > - why is our work not original. Please provide a few peer work in BCSD domain.
> > >
> > > Thanks again for your time.

---

### Decision · Program_Chairs · 2025-09-17

**Decision:**

Accept (poster)

**Comment:**

This paper tackles binary code similarity detection (BCSD) and proposes EBM (Effective Binary Matching), a finetuning process combining multiple techniques: data augmentation with additional metadata tokens, translation learning, embedding training with LLM2Vec, and contrastive learning with a variation of GTE loss. Experiments show that this outperforms multiple LLM-based and non-LLM-based baselines for BCSD, and an ablation study shows that each component of the finetuning pipeline contributes substantially to the overall performance.

Reviewers mentioned the following strengths:
* The method is easy to understand and conceptually well-justified
* The experimental setup is thorough: baselines include multiple non-LLM approaches, finetuned generic LLMs, and LLM approaches specific to BCSD; experiments are performed on multiple datasets; an ablation study quantifies the effect of each component of EBM
* The empirical results are strong with EBM clearly outperforming all of the other compared approaches. In particular, Qwen2.5-Coder-**0.5B** finetuned with EBM outperforms the much larger Qwen2.5-Coder-**1.5B** finetuned on the same data with only contrastive learning.

After discussion, some remaining weaknesses include:
* The EBM finetuning approach is only applied to one model, so it is not completely clear how much the approach would generalize to different base models (but there are also no concrete reasons to suspect poor generalization)
* The downstream applications of BCSD, which include vulnerability and malware detection, could use more thorough discussion or experiments to highlight the practical usefulness of the method
* Some parts of the writing are unclear, and there are formatting issues (improper in-text citation format, sub-subsections without subsections) which make reading more difficult

I believe the strong experiments and results outweigh the minor weaknesses, and four of five reviewers are on the side of accept. Thus, I recommend to _accept_ this paper.

_I strongly urge the authors to add needed clarifications to the text and improve the formatting in the next revision._